# Electrochemical Characterization of Aromatic Molecules with 1,4-Diaza Groups for Flow Battery Applications

**DOI:** 10.3390/molecules26082227

**Published:** 2021-04-12

**Authors:** Alexandros Pasadakis-Kavounis, Vanessa Baj, Johan Hjelm

**Affiliations:** Department of Energy Conversion and Storage, Technical University of Denmark, 2800 Kgs. Lyngby, Denmark; alpaka@dtu.dk (A.P.-K.); vanbaj@dtu.dk (V.B.)

**Keywords:** aqueous organic flow batteries, materials, electrochemistry, NMR

## Abstract

The aqueous redox flow battery is a promising technology for large-scale low cost energy storage. The rich possibilities for the tailoring of organic molecules and the possibility to discover active materials of lower cost and decreased environmental impact continue to drive research and development of organic compounds suitable for redox flow battery applications. In this work, we focus on the characterization of aromatic molecules with 1,4-diaza groups for flow battery applications. We examine the influence of electron-withdrawing and electron-donating substituents and the effect of the relative position of the substituent(s) on the molecule. We found that electron-withdrawing substituents increased the potential, while electron-donating decreased it, in agreement with expectations. The number of carboxy-groups on the pyrazinic ring was found to have a strong impact on the heterogeneous electron transfer kinetics, with the slowest kinetics observed for pyrazine-2,3,5,6-tetracarboxylic acid. The stability of quinoxaline was investigated by cyclic voltammetry and in a flow cell configuration. Substitution at the 2,3-positions in quinoxaline was found to decrease the capacity fade rate significantly. Furthermore, we demonstrated how molecular aggregation reduces the effective number of electrons involved in the redox process for quinoxalines. This translates to a significant reduction of the achievable volumetric capacity at higher concentrations, yielding values significantly lower than the theoretical capacity. Finally, we demonstrate that such capacity-limiting molecular aggregation may be reduced by introducing flexible side chains with bulky charged groups in order to increase electrostatic repulsion and steric hindrance.

## 1. Introduction

The rapid increase in installed capacity of intermittent renewable energy sources motivates research into sustainable and low cost stationary energy storage technologies. Redox flow batteries (RFBs) are particularly attractive for large-scale energy storage due to the decoupled power and energy inherent in such systems, and the resulting falling cost of energy storage with increasing discharge times. Aqueous organic flow batteries are particularly attractive for large-scale energy storage due to the use of a non-flammable electrolyte and the promise of environmentally-benign components. Mass production costs of the active materials and electrolyte lifetime are the primary barriers for widespread introduction of commercial aqueous organic flow batteries. It has been projected that large scale production of organic active materials for flow batteries can have low production costs [1] but requires relatively few synthetic steps and low cost raw materials [2,3].

Organic molecules offer high tailorability through structural modification using a large variety of functional groups and substitutions. This way, high solubility, suitably placed redox potentials, fast charge transfer kinetics, and chemical stabilities in the order of decades may be achieved [4]. The most common groups of organic molecules studied for aqueous RFB applications include quinones [1,2,5,6,7] and *N*-containing organic heterocycles, such as phthalazine [7], quinoxaline [8], and phenazine [9,10,11,12].

Both pyrazine and quinoxaline are stable aromatic compounds containing two nitrogen atoms in the ring. The lone pairs located on the nitrogen atoms are not delocalized in the aromatic ring and therefore provide a nucleophilic character to the molecule, and a weak basicity (pKa of the conjugate acid in the range 0.60–0.88). Furthermore, in contrast to the energy of the *p* orbitals in benzene, *N*-containing aromatic molecules have lower energy orbitals as a result of the greater electronegativity of the nitrogen atom compared to that of carbon. Lower-energy filled *p* orbitals make the molecule less prone to react with nucleophiles but more reactive towards electrophiles [13]. The electrochemical reduction of pyrazine and quinoxaline is characterized by the transfer of two electrons to yield their 1,4-dihydro-counterparts with a varying number of protons involved depending on the pH of the solution [14,15].

### 1.1. Pyrazine

The simplest N-containing aromatic compound presented in this study is pyrazine **1**.

Pyrazine **1** undergoes reduction to give 1,4-dihydropyrazine **1a** through a coupled two electron, three proton transfer reaction (Figure 1). In cyclic voltammetry, at acidic pH (<2) the two individual one-electron transfer processes are visible as two separated peaks, while the two peaks coalesce into a single peak at higher pH [14]. Swartz and Anson performed a series of staircase voltammetric and polarographic measurements of substituted pyrazines (**1**–**5**) at pH 1 in a 0.1 M perchlorate electrolyte [16]. Under those conditions, the total number of protons and electrons involved were three and two, respectively, for all the substituted pyrazines; the reversibility of the different compounds was affected by the pH and the scan rate used. Electron-withdrawing groups such as in molecules **2** and **3** tended to increase stability and shift the redox potential to more positive values, while electron-donating groups (pyrazines **4** and **5**) showed the opposite behavior [16]. Reduced pyrazine **1** undergoes decomposition in aqueous solution that is acid–base catalyzed with the slowest decomposition rate at pH ~3 [14].

### 1.2. Quinoxaline

Quinoxaline (QUI) **6** is an *N*-containing heterocyclic compound made up of a pyrazine ring fused with a benzene ring. The redox potential is more positive relative to that of pyrazine **1** due to the presence of the second aromatic ring [16]. Similar to pyrazine, the total number of electrons involved in the reduction reaction (**6** to **7**, first step, Figure 2) has been postulated to be two [8], while the number of protons depends on the chemical environment and most notably on the pH of the electrolyte solution [8,15,17]. Additionally, the pH has a significant influence on the stability of the reduced form **7**, as well as the reversibility of the oxidation/reduction process. The reduced form of quinoxaline **7** in mild to acidic pH can have two possible degradation mechanisms. In degradation mechanism **a**, quinoxaline **7** undergoes a Michael addition of a molecule of H_2_O to give molecule **8a**, which is thought to be immediately converted to the non-redox active tautomer **9a** [8,15], whereas the other possible pathway includes a catalytic addition of a proton to molecules **8b**. Hereafter, molecule **8b** tautomerizes to **9b**, which is then irreversibly reduced to non-redox active molecule **10b** [18]. The former is visible in cyclic voltammetry as an additional oxidation peak at a more positive potential relative to that shown by its redox potential [15]. A study of the electrochemistry of QUI **6** dissolved in different electrolytes revealed that the presence of at least 10 mM OH^−^ (regardless of the cation counter-ion) (pH > 11) improved cycling stability and reversibility [8].

QUI **6** shows some promising features for utilization as a negolyte in an alkaline redox flow battery. The estimated solubility in >0.9 M KCl electrolyte was reported to be 4 M [8]. The relatively low redox potential of pyrazine and quinoxaline is an attractive feature for a flow battery negolyte, but the limited stability of the reduced state of these molecules is a critical barrier to the utilization of these materials for energy storage. In this study we set out to investigate the properties of a set of pyrazine and quinoxaline derivatives to shed light on the effect of different functional groups, pH, and substitution patterns on critical parameters, such as redox potential, electrochemical kinetics, and molecular stability.

## 2. Results

### 2.1. Substituted Pyrazines

The introduction of functional group alters the physicochemical and electrochemical properties of pristine pyrazine [16]. The type and number of substituents play major roles in the response [19]. Herein, methyl and carboxylic acid groups were chosen as electron-donating and electron-withdrawing groups, respectively. The effect of the nature and the number of substituents on the pyrazine ring was studied by cyclic voltammetry. The structures of six of the investigated pyrazines are shown in Figure 3. It consists of pristine pyrazine (**1**) and mono- (**10**,**13**), di- (**11**,**14**), and tetra-substituted (**12**,**15**) pyrazines with methyl and carboxylic acid substituents.

A 5 mM solution of each analyte in 0.1 M KOH/0.9 M KCl was prepared and analyzed using voltammetry. Based on the shape of the voltammograms (Figure 4), it was clearly visible that the reversibility of the redox couple was reduced, with increasing number of substituents present, for both types of substituents. This effect was especially pronounced in pyrazine-2,3,5,6-tetracarboxylic acid **15**, where the peak separation grew to almost 2 V, indicating very slow electrochemical kinetics. There are several factors that could affect the electrochemical kinetics and cause the large peak separation; these include steric hindrance, electrostatic factors, hydrogen bonding, intra-molecular effects, and structural reorganization [20,21,22]. In the methyl-substituted pyrazines (**10**–**12**), both the anodic and the cathodic peak were shifted negatively by similar values, indicating that the methyl group only affected the energy level of the pyrazine ring and not its kinetic behavior, since the separation of the two peaks remained the same. The average peak potential dropped by approximately 55 mV per methyl group, similar to the 50 mV reported by Swartz et al. in phosphate citrate buffers at pH < 2 [16]. The expected behavior from an electron-withdrawing group would be to shift the potential to more positive values [16]. This clearly applied to the mono-substituted carboxylic acid pyrazine **13**, wherein both oxidation and reduction peak shifted in the predicted direction. In contrast, in more densely-substituted pyrazines **14** and **15** the heterogeneous kinetics were affected with a dramatic increase in the peak split, indicating a much reduced electron transfer rate, obscuring substituent effects on the redox potential to some extent. Nevertheless a small increase in the redox potential with increasing number of substituents was detected for the carboxy-pyrazines (**13**–**15**) (slope = +44 mV per carboxy-group, see inset in Figure 4b).

### 2.2. Substituted Quinoxalines in Unbuffered Conditions

The potential of pyrazine **1** (and its derivatives **10**–**15**) is well past the thermodynamic hydrogen evolution limit and close to the kinetic hydrogen evolution limit. That is a possible obstacle for use in a RFB system for both safety issues and loss of efficiency. Therefore, a representative system of six differently-substituted quinoxalines (Figure 5) was investigated in the same way as for pyrazines with a special focus on the effect of different functional groups. 

Figure 5 illustrates the resulting voltammograms of solutions of 5 mM analyte dissolved in 0.1 M KOH/0.9 M KCl. The kinetics and the redox potential were affected both by the type and the position (relative to the active ring) of the functional groups. Molecules **20** and **18** have charged functional groups attached to the pyrazine ring and displayed the slowest electrochemical kinetics. When similar groups (e.g., –COOH) were attached further away from the pyrazine ring, it had a significantly smaller effect on the kinetics (**19**). 

Closer inspection of the observed average peak potentials (Table 1) shows that the influence of a functional group on the redox potential depends on its relative position to the quinoxaline core. For example, the measured average peak potential of 2-methyl quinoxaline **16** was 40 mV more negative than the one of 5-methyl quinoxaline **17**.

QUI **6** was further studied in unbuffered electrolyte solutions from pH 3.5 to about 13.5 (Figure 6). At pH 11, the average peak potential slope was equal to 44 mV/pH corresponding to 4 electrons and 3 protons, or 2 electrons and 1.5 protons. As shown from previous studies, the total number of electrons involved at low concentrations of QUI **6** is around two [8]. The slopes of the peak reduction and oxidation potential vs. pH differed significantly, 25 and 63 mV/pH, respectively, indicating that the number of protons involved in the two processes are different. The slope of the reduction potential vs. pH was close to 29.5 mV/pH, which corresponds to two electrons and one proton, while for the oxidation it was close to 59 mV/pH corresponding to an equal number of electrons and protons involved, in this case two electrons and two protons. 

### 2.3. QUI **6** and DSMeQUI **20** in Buffered Solutions

In the absence of buffer, the proton coupled reduction process of QUI **6** to **7** causes an increase in the pH close to the electrode surface due to the removal of protons from the solution. The electrochemical behavior of QUI **6** was also examined in buffered solutions where the pH was practically constant [6]. Britton–Robinson universal buffer solution was used due to the wide range of pH available (2–12), alongside the ease of tuning the pH to the desired value by adding KOH without changing the substrate.

The response of QUI **6** in buffered solutions is depicted in Figure 7. Below pH = 9, a second oxidation peak was visible and it increased in size as the pH decreased. It corresponds to the oxidation of dihydroquinoxaline (**7**) to yield the hydroxyl derivative **8a**, which then may tautomerize to **9a** (**8a** to **9a**, Figure 2) [15]. The degradation process to form **8a** was less pronounced in unbuffered solutions, likely due to the local increase of the pH (near the electrode surface) caused by the reduction of QUI **6**. The appearance of multiple peaks at pH = 1.7 was likely due to a combination of different reasons. The oxidation and reduction peaks split into two separate one-electron peaks at low pH [6]. Additionally, QUI **6** may undergo irreversible electrochemically driven degradation reactions at positive potentials (Figure 2), likely explaining the most positive observed small oxidation peak.

DSMeQUI **20** was also studied at various pHs in buffered electrolyte solutions. The substituents in the 2,3 positions hinders OH addition, and therefore the second oxidation peak, thought to correspond to the OH addition was not observed. DSMeQUI **20** appeared to have the fastest kinetics among all the substituted quinoxalines at pH = 3, as indicated by the small peak separation (red and black points on Figure 8b). At low pH values, multiple redox waves were visible both for QUI **6** and for DSMeQUI **20**, likely due to potential inversion, i.e., when the order of the two one-electron processes changed, resulting in separate peaks. Similar behavior was observed in pyrazine [14,16].

The blue lines in Figure 9 depict the first two cycles of the cyclic voltammetry of QUI **6** at pH = 4.5. Depicted in orange is the voltammogram of DSMeQUI **20** at pH = 2.5. Already in the second scan of the QUI **6** solution, a reduction of the peak current was observed, indicating significant degradation of QUI **6** to the electro-inactive compound **9a**. Despite the pH being lower in the DSMeQUI **20** solution, no second oxidation peak was observed, and the peak current was unchanged over several scans. This indicates that the degradation reaction was not taking place in this molecule, at least not on the experimental time-scale employed here. The increased stability of DSMeQUI **20** in comparison to QUI **6** was also evident in their charge–discharge curves (Appendix A). QUI **6** could be cycled five times at pH 13 before most of the capacity was lost. In contrast, DSMeQUI **20** could be cycled at least 110 times before most of the capacity was lost at the same pH. The solution of QUI **6** after five cycles was analyzed by LC–MS (Appendix A). It revealed the formation of two new compounds with masses of 146.86 gmol^−1^ and 134.84 gmol^−1^, corresponding to the oxidized form of **8a** and **10b**, respectively, supporting the two hypothesized degradation pathways. However, detection of **8a** contradicts the idea that it is immediately converted to its non-redox active tautomer (**9a**). QUI **6** was still present as a major component in the solution based on LC–MS analysis. This indicates that chemical degradation of QUI **6** was not the main reason for the observed capacity fade. No leakage was detected, and no volumetric imbalance in the two tanks was observed. The fast capacity fade may related to other factors, such as crossover and/or gas evolution. A quantitative ^1^H-NMR of pristine and post-test QUI **6** electrolyte confirms that QUI **6** is the main component in the post test electrolyte, but also that approximately half of the QUI **6** was lost during testing, indicating that cross-over constitutes a large part of the capacity fade for this molecule. Detailed information on the flow cell battery test conditions are given in Section 2.4.

### 2.4. Concentration Dependent Response

A 0.1 M DSMeQUI **20** (or QUI **6**) solution in 0.1 M KOH/2 M KCl (anolyte) was inserted in a flow battery single cell and paired with 0.2 M of ferrocyanide and 0.2 M ferricyanide (posolyte). Excess capacity of the posolyte was used to make the negolyte the capacity limiting side, as we were interested in the behavior of the negolyte [22]. Carbon cloth (ELAT-H) was used as electrodes and mPBI (10 μm thickness) as a membrane to separate the cathode from the anode. The electrodes were heated for 24 h at 400 °C in air prior to test. A loss of approximately 20% capacity was observed immediately from the first cycle (Appendix A). The negolyte solution was examined by ^1^H-NMR and compared with the pristine solution of DSMeQUI **20** (Appendix A). A new set of peaks arose in the aromatic region, showing the presence of an additional molecule, which could be a degradation product or DSMeQUI **20** in a different protonation state. The integration of the detected new peaks in the NMR spectrum did not completely match a 20% loss in capacity but indicated that the new product corresponds to about 16% and the remainder (84%) was DSMeQUI **20**. A study of dimerization of anthraquinonedisulfonate (2,7-AQDS) presented by Carney et al. [23] showed that aggregation (dimerization) could reduce the effective number electrons of the redox reaction. A non-integer effective number of electrons involved in the QUI **6** redox reaction was reported in a voltammetric investigation by Milshtein et al. [8]. In order to obtain an indication of whether the discrepancy between the theoretical and the observed initial cycle capacity may be related to molecular aggregation, we examined the concentration dependence of the peak currents observed in cyclic voltammetry of DSMeQUI **20**. The peak current density observed in cyclic voltammetry depends on the number of electrons involved, the concentration, and on the diffusion coefficient of the species [24]. A series of cyclic voltammetry measurements were therefore conducted at concentrations of DSMeQUI **20** from 1 to 250 mM (Appendix A). A drop of the concentration (C) and the scan rate (ν) normalized cathodic peak current (I_p_/(ν^1/2^C)) with increasing concentration was observed and indicated that the effective number of electrons decreased with increasing concentration. The same experiments conducted on QUI **6** indicated that the number of electrons involved in the redox reaction also decreased with increasing concentration. Owing to this complication, we were not able to determine the diffusion coefficient and the effective number of electrons via cyclic voltammetry, as we had one unknown too many. Instead, we utilized low-field NMR spectroscopy to investigate the possible aggregation of QUI **6** (Appendix A) and DSMeQUI **20** (Figure 10). 

Concentration dependent peak shifts observed in NMR spectra were monitored in order to gain more information regarding the aggregation constant and the structure (dimers, trimers, etc.) of the possible aggregates. ^1^H-NMR spectra at different concentration of DSMeQUI **20** (from 1 M to 25 mM in the 0.1 M KOH/0.9 M KCl) were recorded. As shown in Figure 10a, with increasing concentrations of the analyte, there was a shift of the aromatic signals toward lower fields and of the CH_2_ connected to the sulfonates group to higher fields. This is indicative of a fast equilibrium between the monomer and the aggregate structure(s) formed by a π-stacking interaction.

The different δ at the various concentrations for both signals 2,3 and 1 were noted and fitted using both the dimer model for aggregation from Horman and Dreux [25] and the isodesmic model [23] (equations described in the Appendix A and one fitting example is shown in Figure 10b). The model of Horman and Dreux assumes that only dimerization takes place, while the isodesmic model takes into account all the possible poly-aggregates. The same titration was performed also with QUI **6** as analyte to understand the effect of the sulfonate groups on this phenomenon. Additionally, a positively charged trimethylammonium quinoxaline **21** (DNMeQUI) was synthesized and analyzed in the same way to confirm to what extent the aggregation was hindered by electrostatic repulsion (Appendix A). The obtained best-fit values after fitting the two models to the data are tabulated below (Table 2).

The obtained aggregation constants (Appendix A) using the two different models were almost identical, indicating that the dominating aggregate formed was a dimer, and that any higher aggregates were only present at very low concentration. The bulky and charged side groups reduced dimerization likely due to both steric hindrance effects and coulombic repulsion. 

The actual concentration of both monomer (C_monomer_) and dimer (C_dimer_) as a function of the total concentration (C_total_) can be calculated based on the following equations.

(1)$$K\mathrm{agg}=C\mathrm{dimer}/C^{2}\mathrm{monomer}$$

(2)Ctotal=Cmonomer+Cdimer

Assuming that a dimer only yields two electrons (only one monomer active with n = 2, or only 1 electron per monomer), one may estimate the effective number of electrons as a function of total concentration based on the following equation:

(3)neff=2(Cmonomer+Cdimer)/Ctotal

Using Equation (3) and the obtained aggregation constant, the effective number of electrons was computed for the three compounds (**6**, **20**, and **21**) and shown in Figure 11. The impact of the aggregation on the effective number of electrons and the effectiveness of charged side-groups for reducing this problem are illustrated in Figure 11.

## 3. Discussion

Organic molecules present a possibility of use as energy materials in aqueous organic RFB. In this study, we focused on the effect of different functional groups on aromatic 1,4-diaza molecules. 

Firstly, it was shown that electron-withdrawing groups increased the potential of the molecule, while electron donating tended to decrease it. The position of the functional group relative to the active redox center (the pyrazine ring) played a major role as well, as the closer it was, the stronger the effect. Additionally, multiple charged substituent groups (carboxylate) on the pyrazine ring had a strong influence on the heterogeneous electron transfer kinetics, evident from the increased peak splitting. Degradation pathways can be hindered with the correct placement of functional groups, as seen in the case of DSMeQUI **20** compared to QUI **6**. Thus, it is of great importance to investigate and map different degradation processes that can occur to be able to design highly durable compounds. We also identified two degradation products present in electrochemically cycled QUI **6**-containing electrolyte with a pH = 13, corresponding to quinoxaline-2-ol (**8a**) and 1,2,3,4-tetrahydroquinoxaline (**10b**).

Secondly, pH plays a major role both with respect to the heterogeneous rate constant (kinetics) and on the degradation.

Thirdly, the dimerization of relatively planar aromatic organic compounds is a phenomenon that affects the effective number of electrons and thus the accessible capacity of a flow battery. π-stacking was identified as the most likely interaction driving the dimerization. The degree of dimerization was influenced by the functional groups attached due to coulomb repulsion and steric hindrance. ^1^H-NMR was employed to calculate the dimerization constant, similar to previous works [23,24]. Finally, based on the dimerization constant, we were able to estimate the effective number of electrons for the three model compounds, namely QUI **6**, DSMeQUI **20**, and DNMeQUI **21**. At a concentration of 2 M (which is reasonable for flow cell battery application) the effective numbers of electrons for QUI **6**, DSMeQUI **20**, and DNMeQUI **21** were approximately 1.26, 1.48, and 1.8, respectively. These values correspond to 63%, 74%, and 90%, respectively, of the expected theoretical capacity. In this context, we note that we observed significantly lower capacities than predicted by the dimerization equilibrium constant and our assumptions about the electroactivity of the monomers (n = 1 per monomer) that make up the dimers. This is possibly a consequence of degradation, which was relatively rapid even though substitution at the 2,3-position appeared to be an efficient strategy to prevent or slow down addition of water to the pyrazine ring. In the case of QUI **6**, after testing (5 cycles, pH 13) we observed mainly QUI **6** in solution (See Appendix A), while the capacity was nearly zero at the end of the test. Significant cross-over was detected for QUI **6**, but as it cannot explain the whole capacity fade, it indicates that another irreversible reaction may be causing an imbalance in the cell. Future work is aimed at identifying the degradation products of the protected molecule DSMeQUI **20** and at determining to what extent other side-reactions are contributing to the capacity fade observed in such alkaline quinoxaline-based flow batteries.

## 4. Materials and Methods

The current work focused on the electrochemical behavior of substituted pyrazines and quinoxalines under aqueous buffered and unbuffered solutions. Pyrazine 1,2-methylpyrazine **10**, 2,5-dimethylpyrazine **11**, 2,3,5,6-tetramethylpyrazine **12**, pyrazine-2-carboxylic acid **13**, pyrazine-2,5-dicarboxylic acid dihydrate **14**, QUI **6**, quinoxaline-6-carboxylic acid **19**, 2-methylquinoxaline **16**, 5-methylquinoxaline **17**, and 3 hydroxyquinoxaline-2-carboxylic acid **18** were purchased from Sigma-Aldrich (Søborg, Denmark) and used without any further purification, while pyrazine-2,3,5,6-tetracarboxylic **15** acid was synthesized in-house. Sodium quinoxaline-2,3-bismethylsulfonate (DSMeQUI) was synthesized in house by an SN2 type reaction on 2,3 bis(bromomethyl)quinoxaline. The unbuffered solutions were based on KCl, KOH, and 37% HCl as electrolytes, while for the buffered solutions the Britton–Robinson universal buffer was used and adjusted to the desired pH by the addition of solid KOH.

Cyclic voltammetry (CV) and steady state rotating (ring) disk voltammetry (RRDV) were conducted with a CH 760E workstation potentiostat (CH instruments, Inc., Dallas, TX, USA). In both cases, a three electrode configuration was used with the working electrode (WE), being glassy carbon in CV, and a rotating ring disk electrode consisted of glassy carbon (disk) and gold (ring) in RRDV. The counter and reference electrodes in both cases were a coiled platinum wire and an Ag/AgCl (3 M NaCl) electrode, respectively. Battery cycling took place with a Biologic SP-300 potentiostat (Biologic, Seyssinet-Pariset, France). Before cycling, the solution were degassed with Ar to remove dissolved O_2_. Afterwards, Ar was blown over the solution to mitigate further O_2_ ingress.

Nuclear magnetic resonance (NMR) analyses were conducted using a Spinsolve 80 benchtop NMR spectrometer (Magritek, Aachen, Germany). For the titration studies, the samples were analyzed with the conditions reported in the Appendix A.

## Figures and Tables

**Figure 1 molecules-26-02227-f001:**
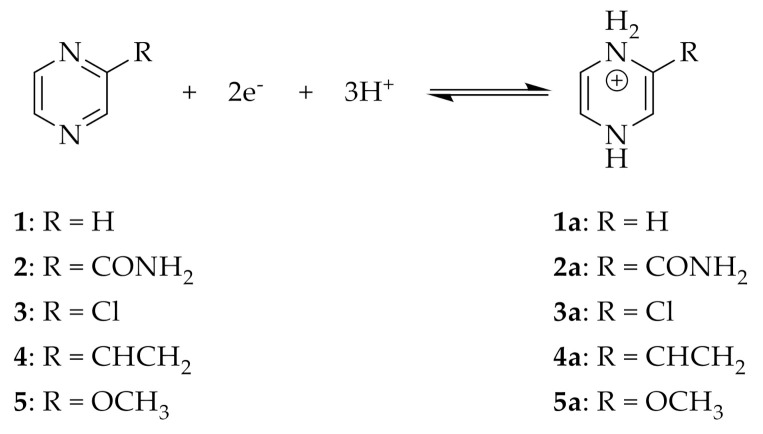
Chemical structures of the different pyrazines studied by Swartz and Anson [16] and the general reduction scheme.

**Figure 2 molecules-26-02227-f002:**
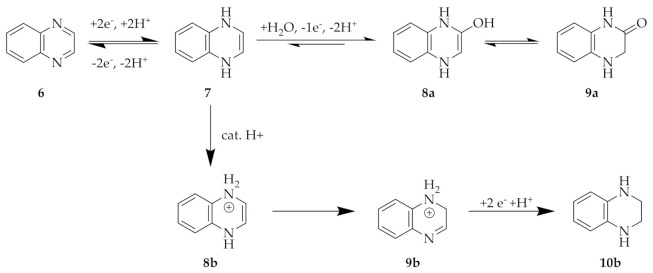
Degradation mechanism of quinoxaline (QUI) **6** in acidic to neutral conditions. (**a**) Formation of quinoxalin-2-ol and/or 1,2,3,4-tetrahydroquinoxaline-2-one [15]. (**b**) Formation of 1,2,3,4-tetrahydroquinoxaline [18].

**Figure 3 molecules-26-02227-f003:**
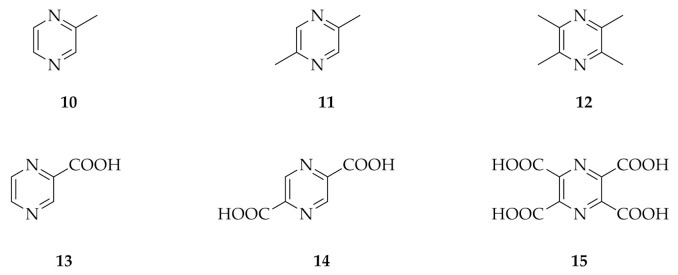
Chemical structures of the different pyrazines studied.

**Figure 4 molecules-26-02227-f004:**
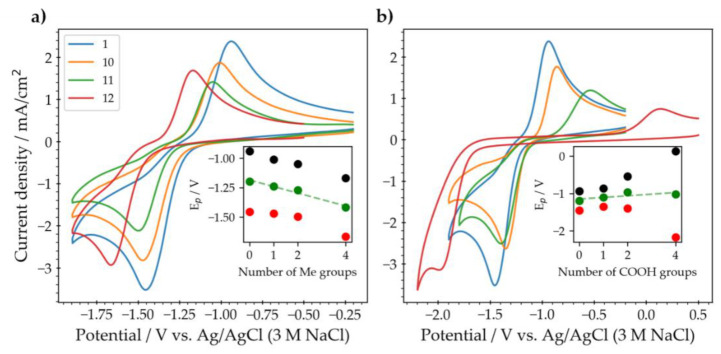
(**a**) Cyclic voltammetry of pyrazine **1** (blue), and mono- **10** (orange), di- **11** (green), and tetra-methyl **12** (red)-substituted pyrazine. (**b**) Cyclic voltammetry of pyrazine **1** (blue) and mono- **13** (orange), di-**14** (green), and tetra-pyrazine-carboxylic acid **15** (red). The black, red, and green circles on the inset graphs correspond to the oxidation, reduction, and average peak potential of each different pyrazine as a function of the number of functional groups. The concentration of each analyte was 5 mM in 0.1 M KOH/0.9 M KCl. The voltammograms were recorded at a scan rate of 250 mV/s using a 3 mm diameter glassy carbon disk electrode.

**Figure 5 molecules-26-02227-f005:**
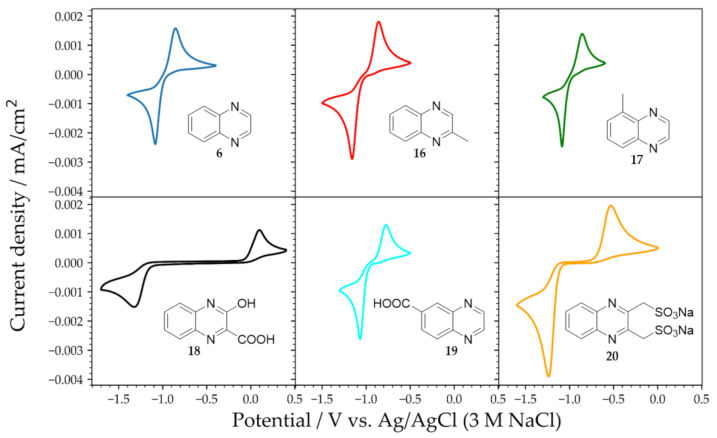
Cyclic voltammetry of 5 mM solutions of quinoxalines **6**, **16**–**20** dissolved in 0.1 M KOH/0.9 M KCl. The voltammograms were recorded at a scan rate of 100 mV/s using a 3 mm diameter glassy carbon disk electrode.

**Figure 6 molecules-26-02227-f006:**
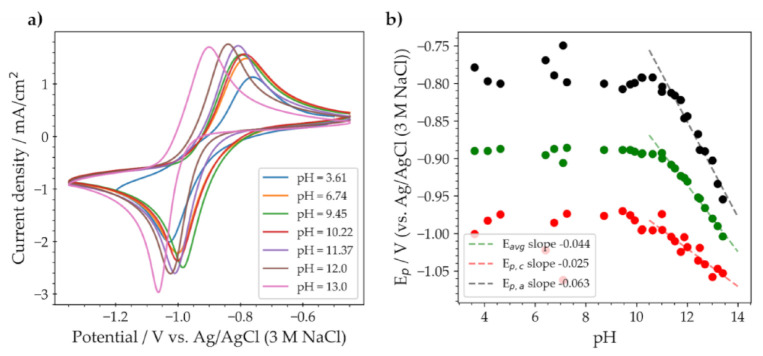
(**a**) Cyclic voltammograms of a 5 mM solution of QUI **6** in unbuffered solution of 1 M KCl at different pHs. (**b**) The resulting potential-pH diagram shows the reduction (red), average (green), and oxidation (black) peak potentials. The voltammograms were recorded at a scan rate of 100 mV/s using a 3 mm diameter glassy carbon disk electrode.

**Figure 7 molecules-26-02227-f007:**
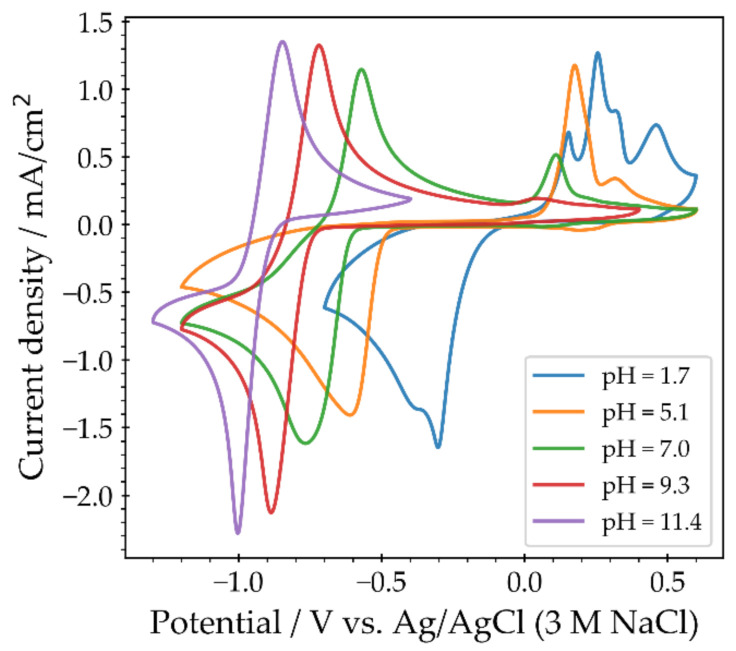
Cyclic voltammetry of pristine quinoxaline on GC in a Britton–Robinson universal buffer solution with added KCl. The pH ranged from 1.7 to 11.4. The voltammograms were recorded at a scan rate of 100 mV/s using a 3 mm diameter glassy carbon disk electrode.

**Figure 8 molecules-26-02227-f008:**
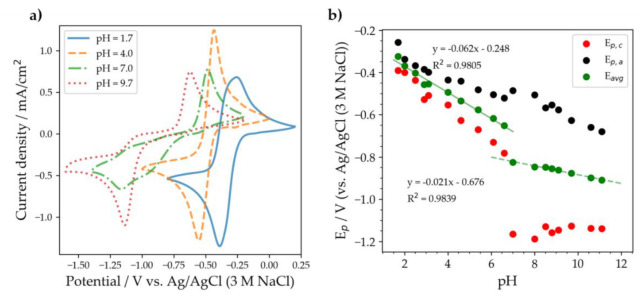
(**a**) Cyclic voltammetry of a 5 mM DSMeQUI **20** solution in a Britton–Robinson universal buffer solution with added KCl at various pHs. The voltammograms were recorded at a scan rate of 100 mV/s using a 3 mm diameter glassy carbon disk electrode. (**b**) The reduction (red) and oxidation (black) peak potential and average peak potential (green) as a function of pH.

**Figure 9 molecules-26-02227-f009:**
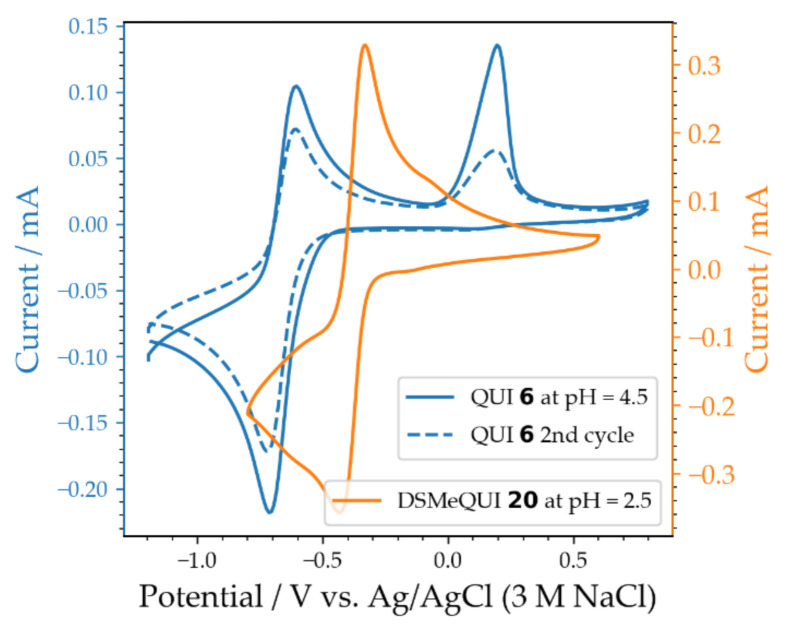
Cyclic voltammetry of 5 mM QUI **6** (blues) and 5 mM DSMeQUI **20** (orange) solutions in Britton–Robinson buffer adjusted to pH 4.5 and pH = 2.5, respectively. The voltammograms were recorded at a scan rate of 100 mV/s using a 3 mm diameter glassy carbon disk electrode.

**Figure 10 molecules-26-02227-f010:**
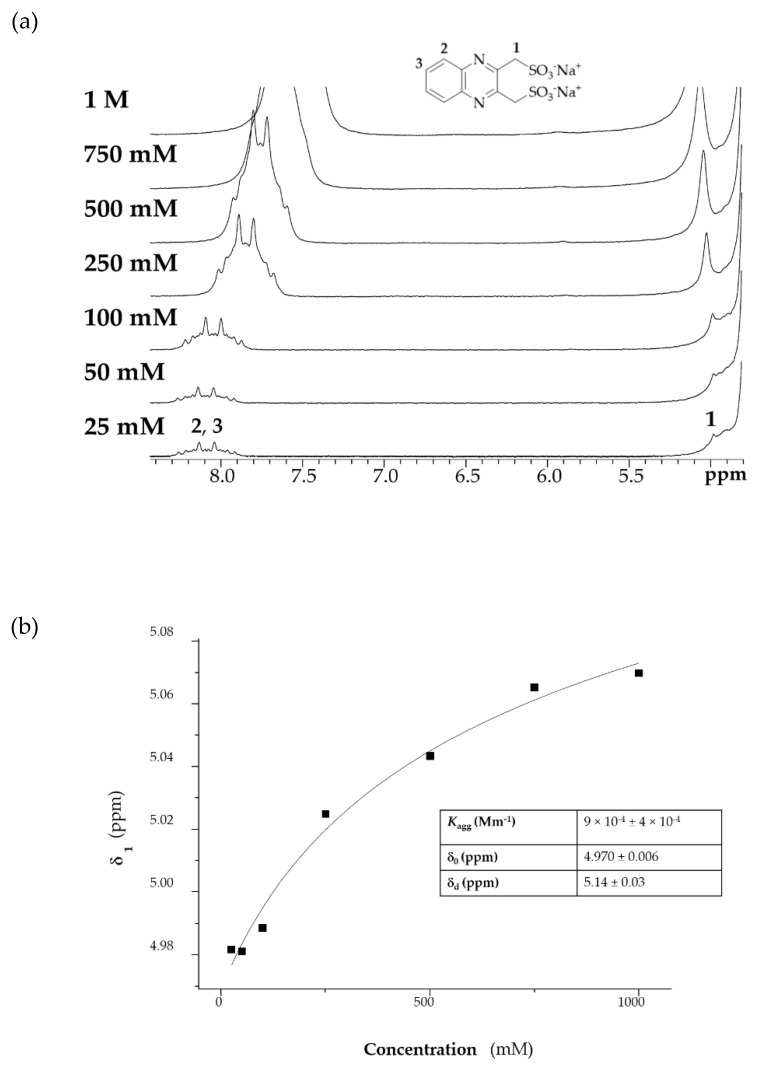
(**a**) ^1^H-NMR spectra with increasing concentration of DSMeQUI **20**. (**b**) Non-linear least squares fitting using the Horman and Dreux model to determine the aggregation constant of DSMeQUI **20**.

**Figure 11 molecules-26-02227-f011:**
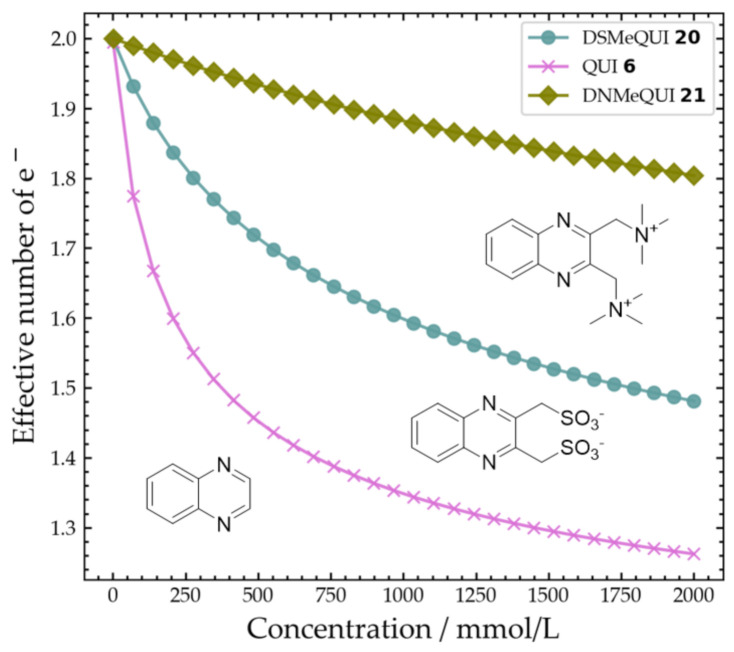
Effective number of electrons (e^−^) predicted as a function of the total concentration of the monomer for the three different quinoxalines (**6**, **20**, and **21**). The predictions are made based on Equations (2) and (3) and obtained aggregation constants.

**Table 1 molecules-26-02227-t001:** Average peak potential of 5 mM solutions of quinoxalines **6**, **16**–**20** dissolved in 0.1 M KOH/0.9 M KCl. All potentials are given versus the Ag/AgCl (3 M NaCl) reference electrode.

Compound	Average Peak Potential (V) Vs. Ag/AgCl (3 M NaCl)
Quinoxaline (QUI) **6**	−0.970
2-Methylquinoxaline **16**	−1.012
5-Methylquinoxaline **17**	−0.972
2-Hydroxyquinoxaline-3-carboxylic acid **18**	−0.458
Quinoxaline-6-carboxylic acid **19**	−0.923
Sodium quinoxaline-2,3-diyldimethanesulfonate (DSMeQUI) **20**	−0.890

**Table 2 molecules-26-02227-t002:** Aggregation constants of the three molecules investigated (QUI **6**, DSMeQUI **20**, and DNMeQUI **21**). Both the Horman and Dreux and isodesmic models were used (Appendix A).

Molecule	Kagg (mM^−1^) Horman and Dreux Model	Kagg (mM^−1^) Isodesmic Model
QUI **6**	2.68 (±1.58)	2.73 (±1.21)
DSMeQUI **20**	0.56 (±0.32)	0.73 (±0.35)
DNMeQUI **21**	0.08 (±0.1)	0.09 (±0.14)

## Data Availability

The data for of the voltammograms can be found here https://doi.org/10.5281/zenodo.4670042. The NMR data can be provided upon request.

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
