# Peer review of "Electrochemical Characterization of Aromatic Molecules with 1,4-Diaza Groups for Flow Battery Applications"

_molecules, 2021, doi:10.3390/molecules26082227_

Round 1

Reviewer 1 Report

Overall, the presented article is very interesting, both from the experimental and the theoretical point of views. It is easy to read and comprehend and brings additional information concerning the stability of organic molecules used in aqueous redox flow battery systems, which is a field under development. Some small questions still need to be answered and a few notes are given for some noticed minor mistakes along the text.

Question 1:

Regarding the reaction presented in Figure 1, why are 3 protons needed for the reaction? and how is the electroneutrality and mass conservation of the system conserved? In fact, as it is, there is an excess of positive charge/proton on the left side of the equation that is not compensated elsewhere. According to reference 17, either you use 1e- and 2H+ to obtain a positively charged product of 1,4-dihydropyrazine or 2e- and 3H+ for a positively charged nitrogen atom. 

Question 2:

In Figure 4, concerning the inset graphs, I don’t think that the color code is the same as the one for the cyclic voltammograms (green: 14 and red: 15), so it probably corresponds to the potential of the anodic and cathodic peaks? If so, maybe it should be specified in the title of the figure.

Question 3:

In Figure 6b, I do not think that the presented graph qualifies as a Pourbaix diagram just because it shows the evolution of the potential as a function of the pH. Pourbaix diagrams are generally more complex schemes based on thermodynamical analysis and plots the predominant and stable species of a specific chemical system according to the pH and the potential range.

Same comment for Figure 8b.

Question 4:

In Figure 8a, at high values of pH (red and green curves), additional oxidation and/or reduction waves appear on the voltammograms. On both curves, an oxidation wave appears at a potential of ~ - 1.1 V and on the green curves (pH = 7), a reduction wave at ~ - 0.8 V can be also observed. To what do you attribute these waves? Do you think that for the DSMeQUI 20 degradation processes can occur at high pHs, as it was observed for QUI 6 at low pHs? 

Question 5:

In lines 229 to 234, the loss of capacity during the cycling of the QUI 6 in the reactor is attributed mainly to the formation of 2 degradation compounds. However, in the supporting info, an additional information is given: “LC-MS analysis showed that 88% of quinoxaline was still present in solution implying that complete degradation of quinoxaline was not the main reason of the battery failure”, and a part of the failure is attributed to the imbalance between the posolyte and the negolyte (understandable). Wouldn’t be interesting to incorporate this small information to the main article?

Question 6: (in supporting info)

Concerning the results obtained in Fig. S6: the supporting electrolyte concentration is indicated to be different as a function of the active species concentration, to “reduce migration effects”. However, since the resulting current is a combination of migration and diffusion (in the absence of convection), did you take that into account during the analysis of the results? In other words, was the supporting electrolyte concentration chosen in a way that the migration contribution to the current is identical for all the active species concentration? And why wasn’t the highest concentration of KOH/KCl used for all of the tested solutions?

Note 1: line 10 correct “todrive” à “to drive”

Note 2: check lines 116 and 177 to rectify the “error reference not found”

Note 3: lines 117 and 118, maybe put the numbers of the corresponding molecules between brackets to avoid confusion

Note 4: in the title of Fig. 7, the pH range is from 1.7 to 11.4 not 12, that is according to the legend placed on the figure

Note 5: In the supporting info, part 1.5.2, be sure to match the designation of the graphs in Fig. 8 with their description in the text (left and right does not fit with the aimed graph)

Note 6: line 352 correct “pyrazine ting” à “pyrazine ring”

Author Response

Author responses highlighted in pale yellow

Question 1:

Regarding the reaction presented in Figure 1, why are 3 protons needed for the reaction? and how is the electroneutrality and mass conservation of the system conserved? In fact, as it is, there is an excess of positive charge/proton on the left side of the equation that is not compensated elsewhere. According to reference 17, either you use 1e- and 2H+ to obtain a positively charged product of 1,4-dihydropyrazine or 2e- and 3H+ for a positively charged nitrogen atom.

We thank the reviewer for spotting this error, and have chosen to show the balanced reaction with 2e- and 3H+, and a positive charge on the nitrogen in the reduced form, corresponding to the findings in ref 17 at pH 1.

Question 2:

In Figure 4, concerning the inset graphs, I don’t think that the color code is the same as the one for the cyclic voltammograms (green: 14 and red: 15), so it probably corresponds to the potential of the anodic and cathodic peaks? If so, maybe it should be specified in the title of the figure.

The caption of Figure 4 was edited in order to be more specific. We have added the following sentence: “The black, red and green circles on the inset graphs correspond to the oxidation, reduction and average peak potential of each different pyrazine as a function of the number of functional groups.”

Question 3:

In Figure 6b, I do not think that the presented graph qualifies as a Pourbaix diagram just because it shows the evolution of the potential as a function of the pH. Pourbaix diagrams are generally more complex schemes based on thermodynamical analysis and plots the predominant and stable species of a specific chemical system according to the pH and the potential range.

Same comment for Figure 8b.

We agree with the reviewer that it doesn’t fully qualify as a Pourbaix diagram, therefore we have changed the captions of Figures 6b and 8b to use “potential – pH diagram” instead of “Pourbaix diagram”.

Question 4:

In Figure 8a, at high values of pH (red and green curves), additional oxidation and/or reduction waves appear on the voltammograms. On both curves, an oxidation wave appears at a potential of ~ - 1.1 V and on the green curves (pH = 7), a reduction wave at ~ - 0.8 V can be also observed. To what do you attribute these waves? Do you think that for the DSMeQUI 20 degradation processes can occur at high pHs, as it was observed for QUI 6 at low pHs?

The second oxidation and reduction peaks, observed at pH 1.7, are most probably caused by potential inversion (i.e. when the order of the two one-electron processes changes resulting in separate peaks). Similar observation have been made with pyrazine (see for example Swartz and Anson), wherein at values less than pH 2 two pairs of peaks appeared for both reduction and oxidation, which corresponded to one electron transfer each. The origin of the additional reduction wave at ~-0.8 V at pH 7 for DSMeQUI is not clear. Possibly potential inversion in this case too, as we only observe a single cathodic wave with a higher peak current already at pH 9.7. We think that degradation of both 6 and 20 also takes place at high pH (but more slowly), we detect degradation products in post-test solutions of both compounds.

Question 5:

In lines 229 to 234, the loss of capacity during the cycling of the QUI 6 in the reactor is attributed mainly to the formation of 2 degradation compounds. However, in the supporting info, an additional information is given: “LC-MS analysis showed that 88% of quinoxaline was still present in solution implying that complete degradation of quinoxaline was not the main reason of the battery failure”, and a part of the failure is attributed to the imbalance between the posolyte and the negolyte (understandable). Wouldn’t be interesting to incorporate this small information to the main article?

The reviewer is correct. However, going through our notes, we found that the LC-MS experiment was made in a qualitative manner. We have not found such an analysis published before so we thought it is important to identify the correct mass of the oxidized compound 8. We have removed the sentence regarding the 88% of quinoxaline from the supporting info since the method was not set up for a quantitative analysis. The percentages shown on the chromatogram in Figure S10 are taken automatically by the instrument over a wide range of wavelengths (210-500 nm) and without any normalization or baseline correction which would be needed for a fully quantitative conclusion. Furthermore we don´t know if the extinction coefficients of QUI 6, the degraded compounds 8a/9a, and the electroinactive 10b are similar and therefore if the integration of the corresponding peaks are directly comparable. So, in addition of removing the sentence in the Supporting Info, we have added clarification about this in the caption of Figure S10, and we have also added the following sentences regarding the analysis in the supporting information:

The instrument method was only qualitative with respect to the amounts of the compounds, as wide band UV-Vis detection (210 – 500 nm) was used and the individual components’ extinction coefficients are not established. Therefore the peak areas only qualitatively indicate the amount of each compound, and cannot be taken as mole fractions. Assuming the extinction coefficients of the degradation product (8a/9a) and the electroinactive sideproduct 10b are not dramatically altered from that of quinoxaline (6) then quinoxaline was the majority component after test, even though the capacity had faded to nearly 0%.”

In the manuscript we have added the following sentence instead as we have NMR data to support the (qualitative) LC-MS observations:

A quantitative 1H-NMR of pristine and post-test QUI 6 electrolyte confirms that QUI 6 is the main component in the post test electrolyte but also that approximately half of the QUI 6 was lost during testing, indicating that cross-over constitutes a large part of the capacity fade for this molecule. ”

Question 6: (in supporting info)

Concerning the results obtained in Fig. S6: the supporting electrolyte concentration is indicated to be different as a function of the active species concentration, to “reduce migration effects”. However, since the resulting current is a combination of migration and diffusion (in the absence of convection), did you take that into account during the analysis of the results? In other words, was the supporting electrolyte concentration chosen in a way that the migration contribution to the current is identical for all the active species concentration? And why wasn’t the highest concentration of KOH/KCl used for all of the tested solutions?

The reviewer is correct, and they way it was written: “reduce migration effects” was a poor description of the situation. The concentration of the supporting electrolyte was increased in order to maintain at least 1/10 ratio of ionic strength of analyte/supporting electrolyte to make sure diffusion currents dominate the response. Here it is worth noting that we used the cathodic peak current, for which the reactant is the neutral quinoxaline (6), and therefore not subject to migration. But the primary reason the supporting electrolyte concentration was increased was to reduce IR drop, as it became significant at the highest analyte concentrations. We only use the peak current change with concentration as a qualitative indicator for an effective reduction in the number of electrons and thus a sign of capacity limiting dimerization/aggregation. The diffusion coefficient should also decrease upon aggregation which would further lower the peak currents. Due to this mix of effects we abstain from a quantitative analysis, and believe it is a fair indicator despite possible minor errors from changes in viscosity, diffusion coefficients, or influence by migration. The reason for not using the highest KOH/KCl concentration for all solutions was that we wanted to stay close to the conditions used in most other measurements.

Note 1: line 10 correct “todrive” à “to drive” done – see note in manuscript

Note 2: check lines 116 and 177 to rectify the “error reference not found” We have not found these errors in the manuscript so no action was taken.

Note 3: lines 117 and 118, maybe put the numbers of the corresponding molecules between brackets to avoid confusion done – see note in manuscript

Note 4: in the title of Fig. 7, the pH range is from 1.7 to 11.4 not 12, that is according to the legend placed on the figure done – see note in manuscript

Note 5: In the supporting info, part 1.5.2, be sure to match the designation of the graphs in Fig. 8 with their description in the text (left and right does not fit with the aimed graph) done – see note in supporting information

Note 6: line 352 correct “pyrazine ting” à “pyrazine ring” done – see note in manuscript

Reviewer 2 Report

In this manuscript, the authors report the electrochemical characterization of aromatic molecules with 2 1,4-diaza groups for flow battery application. The influences of electron-withdrawing or donating substituents, pH values and the dimerization of aromatic organic compounds on potential, kinetics and degradation are discussed. Generally, the quality of this manuscript is high and it is of importance to the research society. This reviewer would recommend it to be accepted by this journal after modifying some minor issues as listed below.

  1. Some descriptions need to be unified in the paper. For example, “dimerization” is more accurate than “aggregation” when referring to aromatic molecules.
  2. In Section 2.2 and Section 2.3, the selection of buffered solution should be explained in detail.
  3. The layout and quality of Figure 10 need to be improved.
  4. Ref. 27 should be “CXCVIII.—Universal buffer solutions and the dissociation constant of veronal”. In addition, is it necessary to cite a reference published in 1931 without a better choice?

Author Response

Author responses highlighted in pale yellow.

We have edited the language where we have found problems/typos/mispelt words.

  • Some descriptions need to be unified in the paper. For example, “dimerization” is more accurate than “aggregation” when referring to aromatic molecules.

Here we have used dimerization for cases where our data indicate that it is primarily dimerization that takes place, and aggregation (the general term) for cases where we are less certain or when the data indicates that poly-aggregates (trimers, tetramers, …) are also formed. We have not taken further action on this but appreciate the point and in general we agree with unified terminology as far as possible.

  • In Section 2.2 and Section 2.3, the selection of buffered solution should be explained in detail.

We have clarified the selection of buffer solution and added the following sentence to section 2.3: “Britton-Robinson universal buffer solution was used due to the wide range of pH available (2-12), alongside the ease of tuning the pH to the desired value by adding KOH without changing the substrate.”

  • The layout and quality of Figure 10 need to be improved.

We agree and have included a new figure that gives more space to both plots included in the figure, and have improved the resolution of the used image file.

  • Ref. 27 should be “CXCVIII.—Universal buffer solutions and the dissociation constant of veronal”. In addition, is it necessary to cite a reference published in 1931 without a better choice?

We thank the reviewer for the correction. The reference has been updated accordingly. We have not come up with a better choice but gratefully accept suggestions. We don’t feel strongly about this reference, admittedly it is old, but appears easily accessible online via the pubs.rsc.org website.